# Mitochondrial dysfunction rapidly modulates the abundance and thermal stability of cellular proteins

Carina Groh[1], Per Haberkant[2], Frank Stein[2], Sebastian Filbeck[3], Stefan Pfeffer[3], Mikhail M Savitski[2], Felix Boos[1], Johannes M Herrmann[1]

Cellular functionality relies on a well-balanced, but highly dynamic proteome. Dysfunction of mitochondrial protein import leads to the cytosolic accumulation of mitochondrial precursor proteins which compromise cellular proteostasis and trigger a mitoprotein-induced stress response. To dissect the effects of mitochondrial dysfunction on the cellular proteome as a whole, we developed pre-post thermal proteome profiling. This multiplexed time-resolved proteome-wide thermal stability profiling approach with isobaric peptide tags in combination with a pulsed SILAC labelling elucidated dynamic proteostasis changes in several dimensions: In addition to adaptations in protein abundance, we observed rapid modulations of the thermal stability of individual cellular proteins. Different functional groups of proteins showed characteristic response patterns and reacted with group-specific kinetics, allowing the identification of functional modules that are relevant for mitoprotein-induced stress. Thus, our new pre-post thermal proteome profiling approach uncovered a complex response network that orchestrates proteome homeostasis in eukaryotic cells by time-controlled adaptations of the abundance and the conformation of proteins.

## Introduction

The eukaryotic proteome consists of thousands of different proteins. Changes in the developmental, environmental or metabolic conditions can induce a considerable remodeling of the proteome. Even seemingly small changes can have pronounced effects. For example, the replacement of the carbon source glucose by galactose in yeast cultures changes the expression of more than 25% of all genes at least twofold, even though galactose catabolism per se requires only the function of five additional enzymes (Ronen & Botstein, 2006). By complex remodeling of the proteome network, cells apparently strive for a maximum of functional performance

and stability, but the molecular mechanisms underlying these transitions are only poorly understood.

Protein homeostasis (proteostasis) depends not only on the abundance of each cellular protein but also on their specific folding, membrane topology, cellular location, and interaction partners and on diverse posttranslational modifications. An elaborate quality control system, also termed proteostasis network, regulates the synthesis, folding, transport, and degradation of proteins (Sala et al, 2017; Hipp et al, 2019; Elsasser et al, 2022 Preprint). It relies on molecular chaperones that facilitate and stabilize protein folding (Hartl et al, 2011) and the ubiquitin–proteasome system (UPS) to recognize and remove damaged or surplus proteins (Dikic, 2017). In addition, a multitude of factors controls the spatial organization of the proteome, either by facilitating protein insertion into and translocation across membranes (Wickner & Schekman, 2005; Wu & Rapoport, 2021) or by concentrating proteins in membrane-less condensates (Sontag et al, 2017; Alberti & Hyman, 2021).

When facing a proteostasis challenge, cells activate stress response pathways that alter the composition of the proteome through changes in protein synthesis or degradation. These changes in protein abundance under different conditions can be determined at high resolution by mass spectrometry (Aebersold & Mann, 2016). However, changes in protein abundance cannot reflect the functional state of a protein, which often depends on its folding state, posttranslational modifications or binding partners. These features remain hidden in classical expression proteomics studies. This is particularly true when examining short timescales, such as the first hours after cells face a proteotoxic challenge. We therefore sought to establish a new method that provides time-resolved and proteome-wide insights into the functional state of proteins under dynamically changing stress conditions. Thermal proteome profiling (TPP) makes use of the cellular thermal shift assay (Martinez Molina et al, 2013) and represents the multiplexed quantification of non-denatured protein fractions as function of temperature (Savitski et al, 2014; Franken et al, 2015; Mateus et al, 2020). Thus, TPP provides a proxy for the thermal stability of each protein in vivo. This powerful strategy is not only perfectly suited for the detection of specific subtle changes, for example, to identify an

---

[1]Cell Biology, University of Kaiserslautern, Kaiserslautern, Germany; [2]Proteomics Core Facility, EMBL Heidelberg, Heidelberg, Germany; [3]ZMBH, University Heidelberg, Heidelberg, Germany

Correspondence: hannes.herrmann@biologie.uni-kl.de; fboos@stanford.edu
Felix Boos's present address is Department of Genetics, Stanford University, Stanford, CA, USA

intracellular binding site of a chemical compound (Mateus et al, 2022), but also to study global changes such as the proteome-wide effects of a specific posttranslational modification (Vieitez et al, 2022). On the other hand, a pulsed labelling approach through the replacement of amino acids by isotopes of different masses in a growing culture (pulsed stable isotope labelling by amino acids in cell culture, pulsed SILAC) can be used to monitor proteome changes of preexisting and newly synthesized proteins at the same time (Ong et al, 2002; de Godoy et al, 2008). This method proved to be very powerful to determine the import, assembly, and degradation of mitochondrial proteins (Bogenhagen & Haley, 2020; Saladi et al, 2020; Schafer et al, 2022). In this study, we combined, for the first time, pulsed SILAC labelling with a two-dimensional TPP approach (2D-TPP) (Becher et al, 2018) which we termed pre-post-TPP (ppTPP). With this, we were able to monitor, in parallel, the time-sensitive variations in abundance and stability of mature and newly synthesized proteins as adaptation to a specific stress-inducing insult. We refer here to these mature and newly synthesized proteins as "pre" and "post", respectively, owing to their synthesis before and after clogger induction and label shift.

We chose the mitoprotein-induced stress response (Topf et al, 2019; Boos et al, 2020) to evaluate the potential of this method. Mitochondria consist of hundreds of proteins which are synthesized in the cytosol as precursor proteins that are posttranslationally imported into the organelle using the translocase of the outer membrane (TOM complex) as a common entry gate (Chacinska et al, 2009; Araiso et al, 2019). Mitochondrial dysfunction can impair protein import causing the cytosolic accumulation of mitochondrial precursor proteins. Such precursor accumulation may occur transiently during development or upon metabolic adaptations, or as a chronic state during aging and age-related diseases (Bauer & Neupert, 2001; Devi et al, 2006; Li et al, 2010; Cenini et al, 2016; Coyne & Chen, 2018; Franco-Iborra et al, 2018; Tsuboi et al, 2020; Eckl et al, 2021).

Induction of a slowly imported clogger protein from a galactose-inducible promoter competitively inhibits mitochondrial protein import and serves as an elegant model to study the consequences of cytosolic precursor proteins in cells that otherwise contain fully functional mitochondria (Fig 1A) (Weidberg & Amon, 2018; Boos et al,

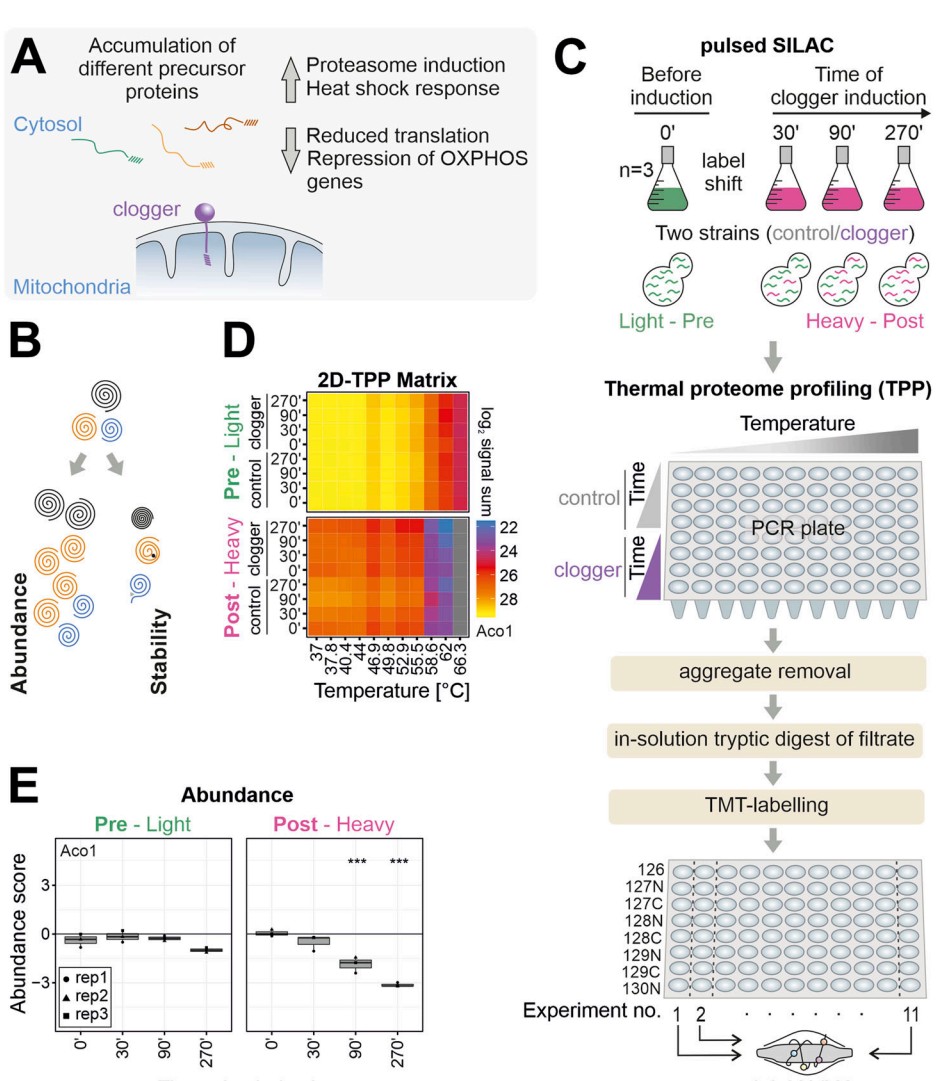

Figure 1. The combination of pulsed SILAC and TPP (ppTPP) allows it to measure stress-induced effects on protein abundance and thermal stability for mature and newly synthesized proteins in one multiplexed approach.

(A) Clogger induction triggers the mitoprotein-induced stress response. (B) The proteome adapts to stress by changing the abundance and thermal stability of proteins. (C) Workflow of the ppTPP approach. For pulsed SILAC experiment, clogger and control cells were grown in a light isotope-containing respiratory medium until the mid-exponential growth phase. Clogger was induced by addition of 0.5% galactose. Simultaneously, the medium was switched from light to heavy isotope-containing medium. Cells were harvested after different timepoints and heated to 11 different temperatures and lysed with NP40 and zymolyase. Nonsoluble proteins were removed by filtration, and the remaining soluble fractions were labelled with isobaric mass tags (TMT) for protein quantification. Different conditions of each temperature were combined in single TMT experiments, which were analyzed by LC–MS/MS. (D) Representation of the 2D-TPP matrix. Shown is the signal sum for different times after clogger induction for proteins synthesized before (upper panel) and after (lower panel) induction in the control and clogger strain for each temperature. Mitochondrial aconitase (Aco1) is shown as the representative example. See Table S1 for details. (E) Pulsed SILAC allows detecting variations for different protein states. Shown is the abundance score (clogger/control) for different times after clogger induction for proteins synthesized before (left panel) or after (right panel) induction. Mitochondrial aconitase (Aco1) is shown as representative example. Asterisks indicating significant p-value (***, <0.001).

2019). Cells tolerate this situation well and react with a multilevel response termed unfolded protein response activated by mistargeting of proteins (Wrobel et al, 2015). This stress response is characterized by adaptive reactions, including the induction of the heat shock response, increased levels of the UPS, and the repression of protein synthesis, in particular, of translation products destined for mitochondria (Wrobel et al, 2015; Boos et al, 2019). But this situation also triggers mitochondria-specific reactions to remove the stalled clogger protein from the TOM complex accomplished by an interplay between a specific AAA-ATPase and proteasome-mediated proteolysis (Weidberg & Amon, 2018; Mårtensson et al, 2019; Shakya et al, 2021).

Mitoprotein-induced stress response seems perfectly suited to explore the potential of ppTPP for several reasons: (i) despite of the immediate and stark response of the cell to clogger induction, cells (and also their mitochondria) stay functional for several hours and maintain their viability; (ii) effects on lipids and metabolites are only secondary through changes in the proteome and, hence, are expected to be much less pronounced than upon heat shock or oxidative stress conditions which directly affect these molecules; (iii) the defined site of action (the clogged mitochondrial TOM complex) allows to distinguish immediate effects of the stress on mitochondrial proteins and secondary effects in the cytosol and other "distant" cellular compartments.

The multidimensional analysis of the ppTPP approach revealed that the cellular proteome reacts on different levels, including various time-dependent changes in protein abundance and thermal stability. We observed that different groups of proteins show highly distinct response patterns and react with subgroup-specific kinetics. Many of these patterns would not have been detectable or discernible by analyzing only abundance, demonstrating the power of ppTPP to discover novel pathways, especially those that act on short timescales.

# Results

### The combination of pulsed SILAC and 2D-TPP enables the proteome-wide profiling of thermal stability and abundance of mature and newly synthesized proteins upon stress induction

The impact of individual aggregation-prone model proteins on cellular fitness and neurodegeneration is well documented. The mitoprotein-induced stress response is arguably more complex as it is elicited by the simultaneous accumulation of hundreds of mitochondrial precursor proteins in the cytosol which exert a complex pattern of effects in the cell (Fig 1A) (Wrobel et al, 2015; Boos et al, 2019; Shakya et al, 2021; Xiao et al, 2021; Schafer et al, 2022). The galactose-controlled induction of the slowly imported model protein cytochrome $b_2$(1–167)-dihydrofolate reductase ($b_2$-DHFR, here also referred to as "clogger") in cells of the yeast *Saccharomyces cerevisiae* proved to serve as a powerful model system for which detailed transcriptome and proteome data are available, demonstrating a complex cellular response with mitochondrion-specific but also generic elements (Weidberg & Amon, 2018; Boos et al, 2019). As a control for the clogger protein, we employed galactose-inducible cytosolic DHFR to address

galactose-induced changes that are independent of mitochondrial protein import (Fig S1A). Expression of the clogger, but not of the DHFR control, slows down cell growth (Fig S1B and C).

To elucidate the clogger-induced adaptations on the abundance and thermal stability (Figs 1B and S1D) of proteins synthesized before or after the onset of clogger expression, we combined 2D-TPP (Becher et al, 2018) with pulsed SILAC in an integrated approach which we termed pre-post-TPP (ppTPP, Figs 1C and S1E). Cells with plasmids for induction of the clogger or the DHFR control were grown to mid-exponential phase on a lactate medium containing $^{14}N_4$-$^{12}C_6$-arginine and $^{14}N_2$-$^{12}C_6$-lysine (light, preinduction). Then, the cells were isolated and transferred to lactate plus galactose medium containing $^{15}N_4$-$^{13}C_6$-arginine and $^{15}N_2$-$^{13}C_6$-lysine (heavy, postinduction). Thus, peptides from proteins that were synthesized before (pre) and after (post) induction can be distinguished on the basis of their masses, thereby allowing to monitor the effects on preexisting (pre) proteins and those on newly synthesized (post) proteins separately.

After 30, 90, and 270 min of expression of clogger or control, aliquots were taken, and the cells were subjected to 3 min incubation at 11 different temperatures ranging from 37°C–66.3°C. After lysis with the mild detergent NP40 and the cell wall-digesting enzyme zymolyase, non-soluble proteins were removed, and NP40-soluble proteins were identified by multiplexed quantitative mass spectrometry based on isobaric tandem mass tag (TMT) labelling. To this end, the individual samples of each time point were TMT-labelled, multiplexed, and measured in a single mass spectrometry-based experiment to determine changes in thermal stability and abundance along the time of clogger induction (Fig 1D).

Using the DHFR strain as control to calculate fold changes (FC), the matrix was condensed to two measures: (i) the abundance (corresponding to the average changes in the two lowest temperatures) and (ii) stability score (corresponding to changes remaining at higher temperatures after correcting for abundance changes), which were calculated using published procedures (Fig S2A) (Franken et al, 2015; Mateus et al, 2020).

The protein abundances resulted in generally stable levels of light-encoded proteins. In contrast, the newly synthesized heavy-encoded proteins showed much stronger protein-specific responses to clogger expression, robustly demonstrating that SILAC labelling reliably distinguished preexisting proteins from those synthesized after clogger expression. For example, newly synthesized aconitase (Aco1), like many mitochondrial proteins, was steadily diminished upon clogger induction compared with control, whereas the levels of pre-existing Aco1 were barely influenced (Fig 1E). The data for all proteins measured are presented in an overview format in Supplemental Data 1.

Thus, the ppTPP procedure allows it to systematically assess the proteome-wide abundance and thermal stability of nascent and mature proteins before and after the onset of stress conditions in one single experiment.

### Stress influences the abundance and the thermal stability of different groups of proteins in characteristic patterns

Do the changes in abundance correlate with those in thermal stability (to which we refer as stability in the following for

**Life Science Alliance**

simplicity)? To answer this question, we calculated pairwise Pearson correlation coefficients for all different samples (Fig 2A) and plotted the changes in stability against those in abundance (Fig S2B). This showed that changes in the stability correlated well with stability changes and changes in abundance with abundance changes throughout the different samples and timepoints. The correlations increased over time of clogger expression both for abundance and for stability, indicating that increasing stress conditions shape a more and more defined and consistent pattern of proteome reorganization (Fig 2A, arrows). However, changes in the abundance and in the stability of proteins represent distinct, non-correlating dimensions. The correlated variables are fundamentally different and show almost no relationship, or only a slight negative correlation for preexisting, but not for newly synthesized proteins.

A heatmap showing the score values of proteins which responded significantly to clogger induction showed the most prevalent changes were those of peptide abundance of newly synthesized proteins (Fig 2B). Many mitochondrial proteins were reduced upon clogger expression (Fig 2B, see proteins marked in blue on left) as a result of their competition with the clogger for mitochondrial import sites. Conversely, proteins of the UPS (Fig 2B, marked in orange on left) showed the opposite trend and were increased upon clogger expression, consistent with the characteristic proteasome induction previously reported on a gene expression level (Boos et al, 2019). Strikingly, the stability changes also showed consistent effects on specific groups of proteins. For example, the stability of many cytosolic ribosomal proteins increased upon the clogger expression (Fig 2B, marked in green on left). Stability, in contrast to abundance, affected both newly synthesized proteins and proteins that were present before the clogger was expressed. Notably, with our SILAC-combined TPP, it is possible to analyze the direct structural and hence functional effect of the clogger on freshly synthesized mitochondrial proteins, which would be hidden when using TPP only (Fig S2C).

Our data demonstrate that abundance and stability changes occur largely independently from each other and are remodeled during mitoprotein-induced stress in a pattern that characteristically and consistently reflects the physiological role of different protein groups.

### Mitoprotein-induced stress strongly diminishes the synthesis of mitochondrial proteins

First, we studied the effect of clogger expression on the abundance of proteins. The signals of light peptides diminished only slightly in the course of 4.5 h, whereas the heavy peptides increased strongly as a result of the ongoing protein synthesis (Fig 3A). Compared with control cells, clogger expression globally reduced the accumulation of new peptides, suggesting a global reduction of protein synthesis, in particular, upon prolonged clogger induction (Figs 3B and S3A). In addition, newly synthesized proteins might also be included into NP40-insoluble aggregates or rapidly degraded in clogger-inducing cells, which likewise would prevent their accumulation. However, the effect of the clogger is very general and not restricted to certain groups of proteins, indicating that the mitoprotein-induced stress slows down cellular translation (Fig 3C and D) consistent with previous observations (Topf et al, 2018, 2019; Boos et al, 2019).

When the synthesis rates (Fig S3B) with and without clogger were compared, different functional protein groups showed characteristic footprints (Figs 3E and S3C): The clogger strongly prevented the accumulation of newly synthesized proteins of the mitochondrial OXPHOS system, either by reducing their synthesis, by inducing their immediate degradation or by stimulating their aggregation (Fig 3E dark blue, Fig S3C and D). On the contrary, the synthesis of proteins of the UPS was strongly increased upon clogger expression (Fig 3E orange, Fig S3C and E).

Subsequently, we tested which individual proteins were most drastically diminished upon clogger expression (Fig 3D). Almost all of the most clogger-reduced proteins were mitochondrial proteins (Fig 3F), suggesting that these membrane proteins are rapidly degraded or sequestered into NP40-insoluble aggregates when their import is compromised. Alternatively, their synthesis might be prevented by some kind of feedback mechanism.

The clogger-induced changes in protein synthesis rates correlate generally well with the changes in mRNA levels (Boos et al, 2019). The correlation was more pronounced for abundant proteins than for lower expressed proteins (Fig S4A–C). However, two groups of proteins deviate from this close correlation: OXPHOS components and other mitochondrial proteins are much more diminished by clogger expression than expected from their mRNA levels, presumably as a consequence of their direct competition with the clogger (Fig 3G). On the other hand, subunits of the proteasome accumulated disproportionately more than expected from their transcript levels. This is in line with recent observations that the unfolded protein response activated by mistargeting of proteins directly promotes proteasome assembly by hyperactivating the biogenesis factors Irc25 and Poc4 (Le Tallec et al, 2007; Kusmierczyk et al, 2008; Wrobel et al, 2015). Ribosomal proteins do not show this distinct separation from the general effect (Fig 3E and G, green). In summary, cells complement the stress-induced changes in gene expression by consistent and protein-specific regulation patterns that operate on a posttranscriptional level (Fig 3H).

### Protein turnover supports the stress-induced remodeling of the proteome

The decline of signal intensities for light peptides over time allowed us to monitor the degradation of proteins that had been synthesized before clogger expression (Figs 4A and S5A). Inhibition of mitochondrial protein import stimulated the protein degradation rate in general to some degree (Fig 4B–D), potentially as a result of the increased proteasome levels. Also, the incorporation into NP40-insoluble aggregates could explain the decline of such preexisting proteins.

The effect of the clogger on the mitochondrial proteins (Fig 4E) goes in line with the general trend, indicating that intra-mitochondrial protein turnover (or aggregation) is accelerated to a specific extent if protein import is blocked. Interestingly, clogger expression increased the selective degradation of specific proteins, including some mitochondrial proteins such as the outer membrane fusion protein mitofuzin (Fzo1) and the MICOS protein Mic12 (Fig 4D and F). Unexpectedly, clogger expression induced the degradation of a number of peroxisomal proteins, indicating a tight crosstalk in the biogenesis of mitochondria and peroxisomes.

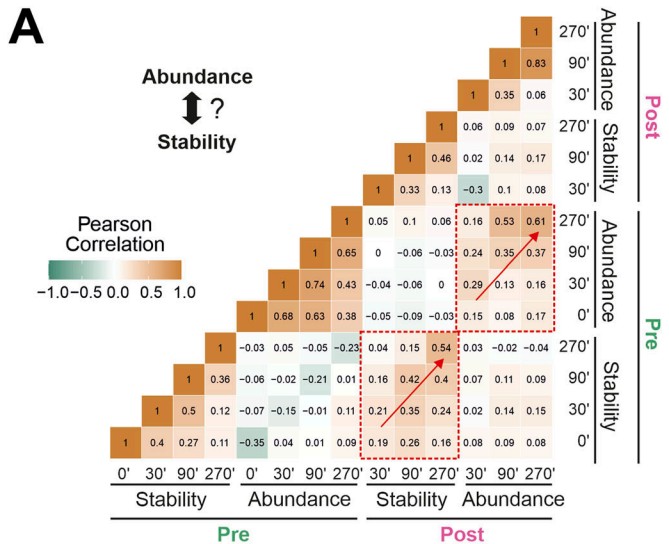

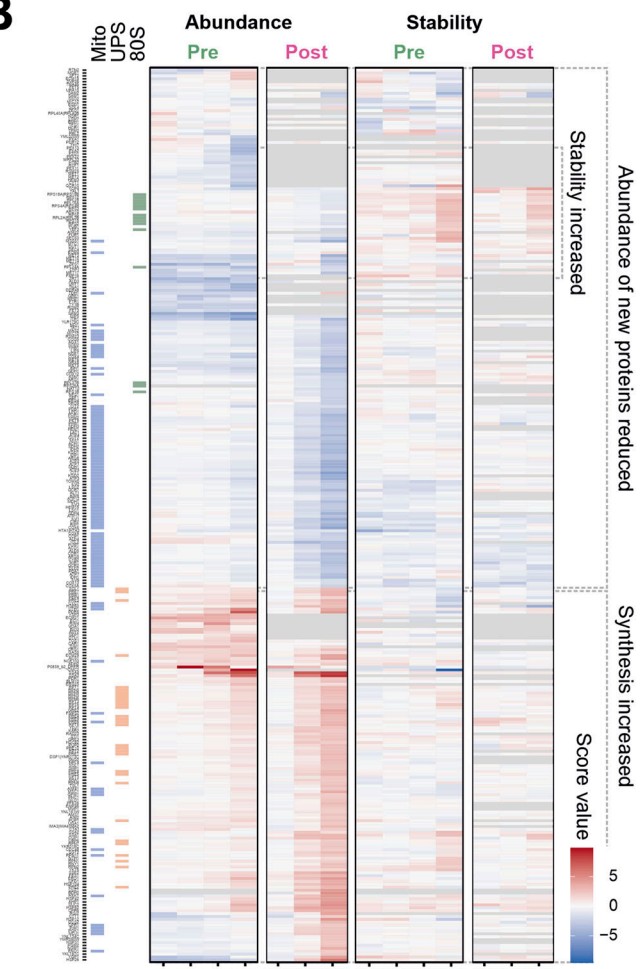

**Figure 2. Stress-induced changes in abundance and stability do not correlate.**
**(A)** Pearson correlation coefficients of all pairwise proteins were calculated using the data generated during ppTPP (clogger versus control). Whereas the different abundance and stability scores correlate with each other, abundance does not or very poorly correlate with stability (see also Fig S2). Red arrows are indicating an increasing correlation with prolonged stress. **(B)** Clustering of

Surprisingly, peroxisomal proteins, especially matrix proteins, were enriched in the class in which synthesis but also degradation was increased, which indicates a strong protein turnover (Fig 4G, high–high).

In many cases, though not always, protein synthesis and degradation show opposed trends (Fig 4G): Clogger expression reduces the synthesis but stimulates the degradation of OXPHOS to ensure a fast and effective protein clearance (Fig S5B). On the contrary, mitoprotein-induced stress increases the production of proteasomes and at the same time, the degradation of UPS components is reduced (Fig S5C). However, ribosomal protein levels and specific OXPHOS proteins are mostly regulated via reduced synthesis (Fig 4H). In summary, the selective degradation of proteins supports the clogger-induced remodeling of the proteome; however, the contribution of proteolysis is much smaller than that of altered protein synthesis.

### Clogger expression changes the thermal stability of proteins of the ribosomal tunnel exit

Because changes in abundance do not reflect structural changes of the proteome, we investigated structural proteome variations as a result of a perturbed proteostasis. The heatmap shown in Fig 2B already revealed that clogger induction leads to a characteristic increase in the thermal stability of proteins of the cytosolic 80S ribosome (Fig 5A). This tendency was obvious for newly synthesized (post) ribosomal proteins and for proteins that had been synthesized before (pre) clogger expression (Figs 5B–D and S6A–C). Thus, inhibition of mitochondrial protein import apparently changes the physical state of the cytosolic translation machinery, influencing thereby both preexisting (i.e., assembled) and newly synthesized ribosomal proteins. Such a feedback of mitochondrial dysfunction on the structural conformation of the cytosolic ribosome had been proposed before from studies that measured the accessibility of cysteine residues on ribosomal subunits (Topf et al, 2018). Mitochondrial function is crucial for efficient protein synthesis in the cytosol (Münch & Harper, 2016), and our observations suggest that cytosolic mitochondrial precursor proteins are directly or indirectly part of the feedback regulation to the ribosome.

Clogger-induced stabilization of mature cytosolic ribosomal proteins was most pronounced for constituents of large subunits such as L4, L18, L25, and L28 (Figs 5E and S6D), which are located in close proximity to the peptide tunnel exit (L25) and contribute to the structural core of the tunnel vestibule (L4, L18, L25) (Figs 5F and S7). Also, many other ribosomal proteins were stabilized, though to a lesser degree (Fig S7). Thus, clogger induction leaves a profound footprint on the structure (and thus the function) of the ribosome which could be a cause or consequence of the reduced translation rates observed when mitochondrial protein import is inhibited.

stability and abundance of significantly changing proteins. Score values of the different samples (clogger versus control) are shown and the distribution of mitochondrial, proteasomal, and ribosomal proteins is indicated on the left. See also Table S1 for the full dataset.

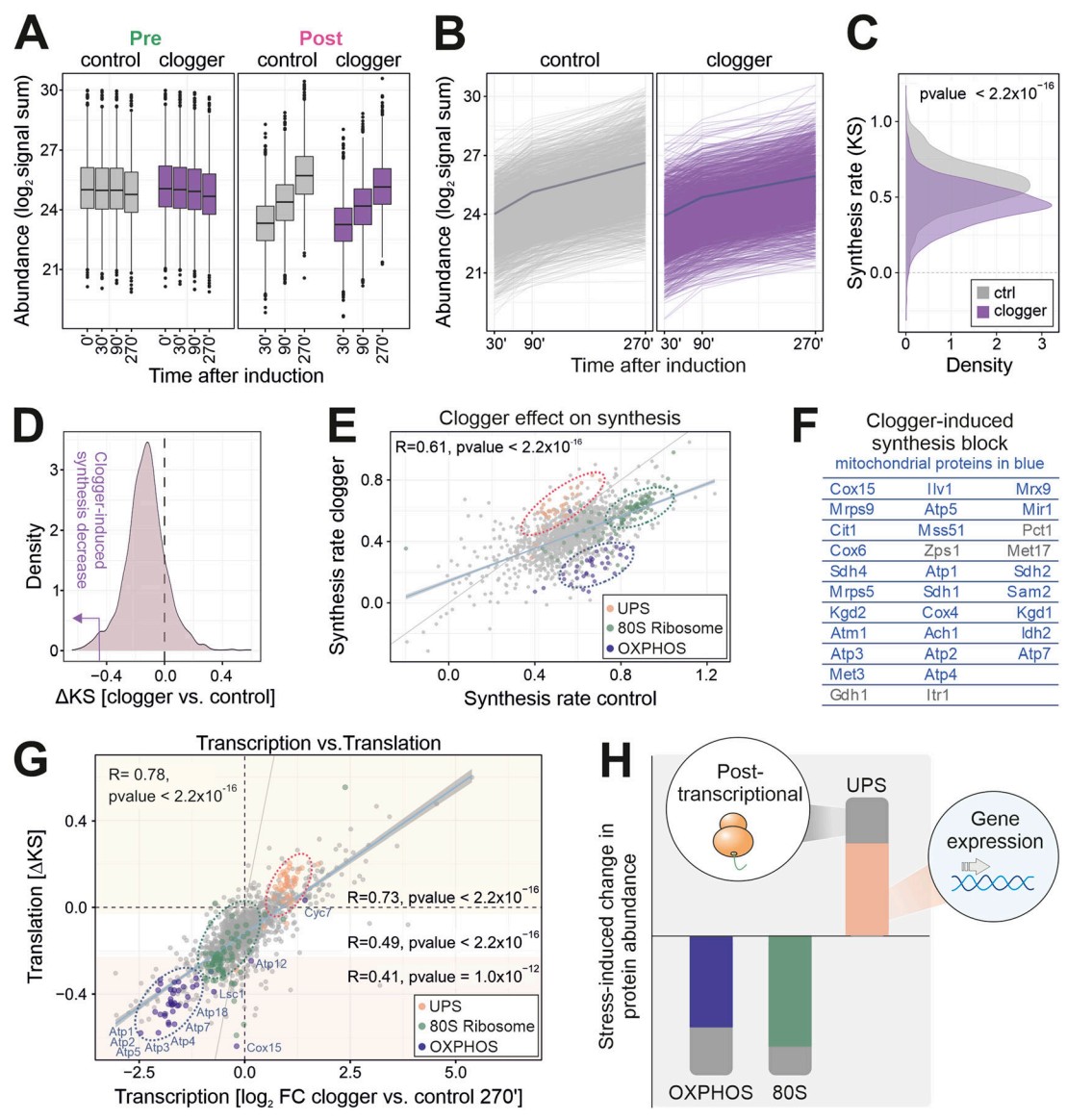

**Figure 3. The stress-response affects protein synthesis by class-specific transcriptional and posttranscriptional adaptations.**
**(A)** Total protein signal sums show the increase of newly synthesized proteins over time (right panel) in contrast to preexisting proteins shown in the left panel. y-axis shows signal intensities of the protein abundance. Line represents the median, box represents the interquartile range, and whiskers of the box represent the 5th and 95th percentiles. **(B)** Signal sums of newly synthesized (heavy) peptides in the absence and presence of clogger. The dark line indicates the mean value over all proteins. **(C)** Density plot showing the synthesis rate (KS) along the 270 min course of clogger induction of stressed and control cells. Synthesis rate was calculated on the basis of heavy peptide intensities. Two-sample Kolmogorov–Smirnov test demonstrates a significant difference between stressed and unstressed cells, revealing lower synthesis rates upon clogger expression. **(D)** Clogger expression reduces the synthesis of most proteins. Density plot of ΔKS, calculated by the ratio of the synthesis rate (KS) of the clogger and control cells. Purple arrow indicating the area used for outlier analysis. **(E)** The synthesis rates of proteins in the presence or absence of clogger were correlated. Proteins of the UPS, the 80S ribosome, and the OXPHOS were labelled. R, Pearson correlation coefficient. **(F)** List of proteins with most pronounced clogger-induced inhibition of synthesis rates. Mitochondrial proteins are indicated in blue. **(G)** The clogger-induced changes in transcription after 270 min (Boos et al, 2019) were compared with those in protein synthesis rates (clogger versus control) showing an overall correlation of 0.78. Pearson correlation coefficient was additionally calculated between transcription and different translation rates: the upper 20%, lower 20%, and middle 60% of the dataset. The names of some proteins with strong clogger-induced synthesis blocks were indicated. Proteins of the UPS, the 80S ribosome, and the OXPHOS were labelled. R, Pearson correlation coefficient. **(H)** Mitoprotein-induced stress results in characteristic changes in protein abundance which are, to a large part, the result of changes in gene expression (shown in color).

## Clogger expression alters the thermal stability of a specific set of mitochondrial proteins

In respect to their stability, mitochondrial proteins were of the group of proteins that was most drastically affected by the clogger

expression. The clogger expression strongly destabilized mitochondrial proteins, in particular, those that were synthesized after (post) clogger induction (Fig 6A). Because these proteins directly compete with the clogger for mitochondrial import sites, this destabilization might arise from

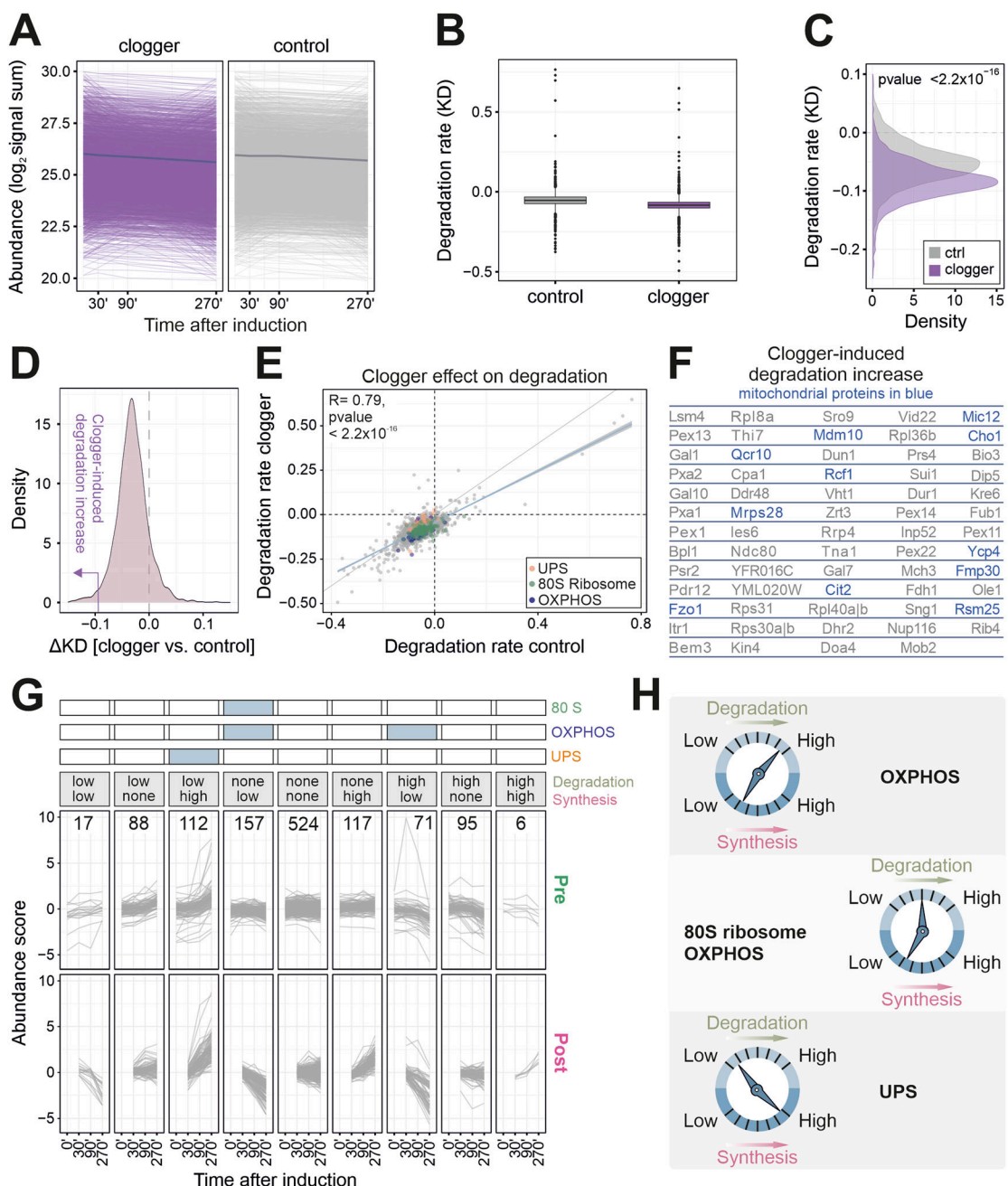

**Figure 4. Competitive inhibition of mitochondrial protein import promotes cellular protein degradation on a large scale.**
**(A)** Signal sums of mature (light) peptides in the presence and absence of clogger. The dark line indicates the mean value over all proteins. **(B)** A box plot demonstrating the degradation rate (KD) along the 270 min course of clogger induction of stressed and control cells. Degradation rate was calculated on the basis of light peptide intensities, revealing higher degradation rates upon clogger expression. **(C)** The degradation rate showing a significantly higher protein degradation of preexisting proteins upon clogger expression. Two-sample Kolmogorov–Smirnov test was used. **(D)** Density plot of ΔKD, calculated by the ratio of the degradation rate (KD) of the clogger and control cells. Purple arrow indicating the area used for outlier analysis. **(E)** The degradation rates of proteins in the presence or absence of clogger were correlated, revealing only minor group-specific effects. Proteins of the UPS, the 80S ribosome, and the OXPHOS were labelled. R, Pearson correlation coefficient. **(F)** Proteins whose degradation was substantially increased upon clogger expression. Mitochondrial proteins are shown in blue. **(G)** Classification for the different categories of abundance regulation. ΔK value of the top or bottom 20% were defined as a considerable change. Numbers are indicating proteins in respective class. Upper part demonstrating enriched protein groups in the respective classes. GO-analysis was performed using the GOrilla tool (Eden et al, 2009), proteins of respective class were used as the target set with all quantified proteins as background. Enrichment is shown with a false discovery rate < 5% and linked to the different classes. See Table S2 for details. **(H)** Degradation and synthesis often show opposite trends, however, the changes in synthesis are much more pronounced than those in degradation.

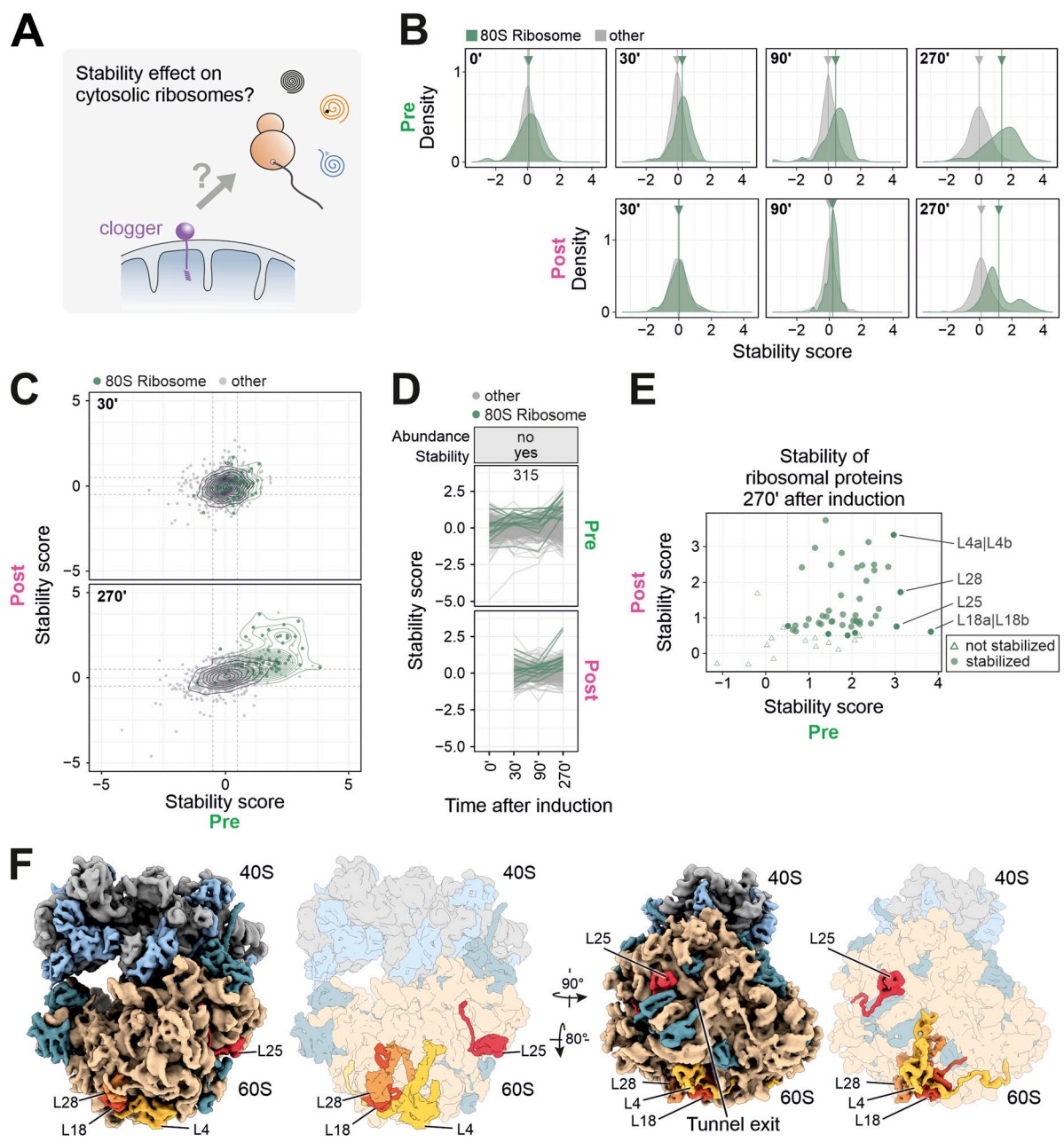

**Figure 5. Clogger expression increases the stability of many ribosomal proteins.**
**(A)** Potential impact of clogger-induced import inhibition on the stability of proteins of cytosolic ribosomes. **(B)** The stability scores of 80S constituents that were synthesized before (pre) or after (post) clogger induction were analyzed. **(C)** Stability changes (clogger verrsus control) of preexisting (x-axis) and newly synthesized proteins (y-axis) were correlated. Green circles show the isobaric distribution of 80S ribosomal proteins, which show an increased stability for both protein pools over time. Black circles show the distribution of the whole proteome. **(D)** Stability scores were plotted of proteins which did not change in abundance but in stability (ΔK value of the top or bottom 20% was defined as a change in abundance; a stability score below −0.5 or above 0.5 was considered as change; the numbers indicate proteins in that category). Traces of cytosolic ribosomal proteins are shown in dark green. See also Fig S6C and Table S3 for details. **(E)** Stability change of cytosolic ribosomal proteins for both preexisting and newly synthesized proteins. Four proteins of the large subunit were indicated which showed the strongest clogger-induced stabilization. **(F)** The positions of ribosomal proteins with considerably increased stability scores are shown. Simulated 3D density of an *S. cerevisiae* 80S ribosome (based on PDB-6Q8Y) (Tesina et al, 2019): Stabilized ribosomal proteins colored in light (SSU) and darker blue (LSU). Highly stabilized mature ribosomal proteins colored individually. Clogger expression increases the thermal stability of many proteins that are in proximity to the tunnel exit of the large ribosomal. See also Fig S7.

their accumulation in the cytosol or their mislocalization to other cellular compartments (Wrobel et al, 2015; Hansen et al, 2018; Weidberg & Amon, 2018; Shakya et al, 2021; Xiao et al, 2021).

In addition to this rather general destabilizing effect, clogger expression caused a particularly strong effect on the stability of a small set of mitochondrial proteins, such as Atp17, Cox5a, Cox13, Hsp60 or Qcr10 (Fig 6B and C). Interestingly, here, even those

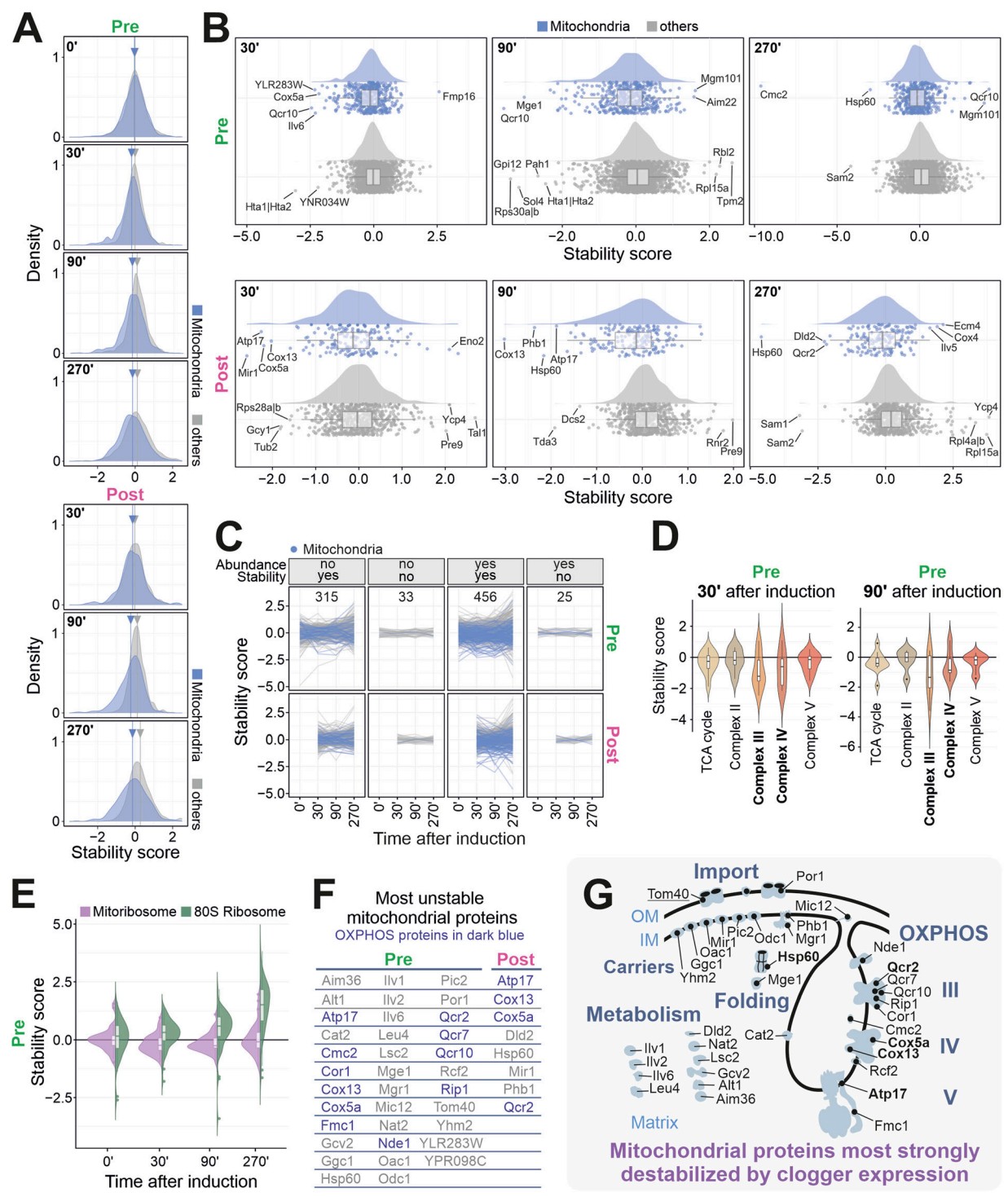

**Figure 6. Inhibition of protein import reduces the thermal stability of many mitochondrial proteins.**
**(A)** Kernel density estimation of stability scores for mitochondrial and nonmitochondrial proteins. Arrow with line indicating the mean values of the stability score, demonstrating the pronounced decrease in stability for mitochondrial proteins upon clogger expression. **(B)** The clogger-induced change in thermal stability was plotted for nonmitochondrial (grey) and mitochondrial (blue) proteins. **(C)** Classification according to Figs 5D and S6C. Mitochondrial proteins are shown in blue. See Table S4 for details. **(D, E)** Range of stability changes shown for different groups of proteins. See also Fig S8. **(F, G)** Shown are mitochondrial proteins which showed the most severe reduction in stability scores upon clogger expression. Unique outlier proteins for every timepoint in each protein pool based on IQR criterion. OXPHOS proteins are indicated in dark purple. Proteins for which both light and heavy peptides showed severe stability changes are labelled in bold.

proteins were destabilized that had been already present before (pre) clogger induction. Thus, the reduced influx of newly synthesized proteins changes the stability of proteins that are already present within the organelle. Stability changes were very pronounced for the enzymes of the OXPHOS system, in particular, for mature complex III and IV (Figs 6D and S8). The stability of Tom40, the channel-forming subunit of the mitochondrial outer membrane translocase, was affected by clogging, as one would expect. We did not detect considerable structural changes for other components of the mitochondrial import machinery detected in our dataset. In general, membrane-embedded proteins were underrepresented among proteins with clogger-induced stability changes.

In contrast to the effects found for the 80S ribosome (see above), the stability of mitochondrial ribosomal proteins was not considerably affected by clogger expression (Fig 6E). Thus, impaired protein import into mitochondria leads to considerable changes in the thermal stability of many mitochondrial proteins, including subunits of specific multi-subunit complexes of the matrix and the inner membrane (Fig 6F). Because these changes affect both newly made and preexisting proteins, it is likely that they indicate a profound remodeling of mitochondrial energy and amino acid metabolism (Fig 6G).

However, not only mitochondrial proteins showed an altered thermal stability upon clogger expression (Fig S9): For example, we noticed profound changes in the stability of the histones H2A, H3, and H4 that form the nucleosome core complex for DNA packaging and many of the coat proteins of clathrin, COP I, and COP II vesicles (Fig S10A). Apparently, clogger-induced stress leads to a complex and comprehensive rewiring of cellular physiology, including processes such as chromatin packaging and intracellular vesicle transport.

### Stability changes as predictor of chaperone occupancy

Are changes in thermal stability suited to predict molecular consequences of clogger expression on a molecular level? To address this question, we had a closer look at the clogger-induced stability changes of molecular chaperones. Clogger expression changed the stability of specific chaperones, in particular, of the cytosolic Hsp70 proteins Ssa1 and Ssa2, and of the mitochondrial Hsp60 chaperonin (Figs 7A and B and S10B). The direct interaction of both chaperone groups with mitochondrial precursor proteins is well established (Cheng et al, 1989; Ostermann et al, 1989; Smith & Yaffe, 1991; Langer et al, 1992; Endo et al, 1996; Hoseini et al, 2016). Moreover, clogger expression changed the stability of the cytosolic disaggregase Hsp104 (Fig S10C) even though the significance values for this protein were not strong; the latter has recently been implicated to be of particular relevance in the context of mitoprotein-induced stress and the formation of MitoStore condensates which stabilize precursor proteins in the cytosol (Nowicka et al, 2021; Krämer et al, 2023). Thus, chaperones do not follow a clear coherent tendency in stability but rather react to clogger expression with protein-specific changes in abundance and stability. This is in contrast to what we had observed for proteins of the proteasome, the 80S ribosome or the OXPHOS system, which mainly respond in concerted group-specific patterns (Fig 7C).

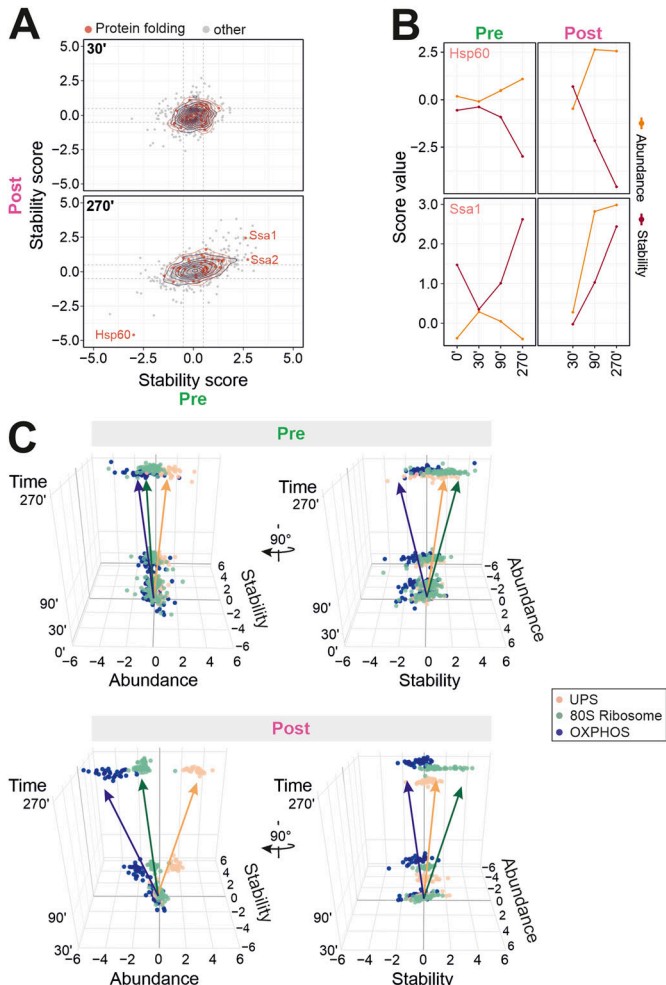

**Figure 7. Cells react to mitoprotein-induced stress with complex adaptations of protein abundance and stability.**
**(A)** Stability changes (clogger versus control) of preexisting (x-axis) and newly synthesized proteins (y-axis) were correlated. Red circles show the isobaric distribution of chaperones. Black circles show the distribution of the whole proteome. **(B)** The stress-induced abundance and stability changes are exemplarily shown for Hsp60 and Ssa1. **(C)** Shown are the distinct functional groups of the UPS, the 80S ribosome, and the OXPHOS system. For every protein of these groups, the stability and the abundance scores at the respective timepoint are shown in a 3D representation. Please note that different groups of proteins follow individual trajectories in the stress response networks.

# Discussion

Cells adapt to different environmental conditions by remodeling their proteomes. These responses can provide resistance to extreme conditions as they arise from different types of stress situations. They are highly complex and affect many if not most proteins to different degrees. Still, on the first glance, cellular responses to stress conditions are seemingly similar (slower growth rates, induction of chaperones, increased proteasomal activity, etc.) even though they need to resolve problems caused by protein-specific or locally defined perturbations. Transcriptomics and proteomics proved to serve as very powerful techniques to study

genome-wide reactions in gene expression and in protein synthesis.

We have developed ppTPP, a strategy that combines pulsed SILAC and 2D-TPP with chemical labelling by TMTs. By using the combination of these powerful methods, ppTPP is able to analyze the abundance and the thermal stability of mature and freshly synthesized proteins of multiple different treatment conditions in a single mass spectrometry-based experiment. Quantification of proteins already synthesized before the onset of stress allows to detect changes in degradation, whereas de novo synthesized proteins allow to detect effects on posttranscriptional regulation. Furthermore, we are able to differentiate structural and hence functional changes between proteins synthesized before and after the onset of stress. We discovered group-specific, and individual protein thermal stability variations which depend inter alia on the maturation and posttranslational modification of proteins and represent an additional layer of global proteome adaptation. In a nutshell, mitoprotein-induced stress generally leads on the level of abundance to (i) reduced protein synthesis including the repression of OXPHOS genes and the induction and increased assembly of the proteasome, (ii) increased protein degradation, and (iii) group-specific synergy between synthesis and degradation. On the level of thermal stability, our data could demonstrate that during stress conditions (i) newly synthesized proteins, prominently including mitochondrial proteins, show a higher vulnerability than non-mitochondrial proteins, (ii) many proteins of the cytosolic ribosome but not of the mitochondrial ribosome increased their stability, (iii) the chaperone network undergoes remodeling to boost the cellular resistance against misfolded proteins, and (iv) diverse functional complexes, including histones and vesicle transport, alter their stability but not their abundance to adapt to surrounding stress conditions. Our results with the higher vulnerability of nascent polypeptides can be explained by their compromised maturation. Because the melting temperature depends on the protein context (Mateus et al, 2020), the detected stability changes can indicate stress-induced alterations of posttranslational modifications, proteolytic cleavages, folding, complex formation or the localization of the protein. Whereas responses on the level of transcription often affect many genes in group-specific patterns (because these genes share regulatory elements such as the heat shock element HSE, the proteasome-associated control element PACE, the pleiotropic drug resistance element PDRE, etc.), posttranscriptional regulation differentially affects individual proteins, for example by posttranslational modifications.

Our ppTPP analysis identified the complex pattern of stability changes in many individual proteins. These changes were independent of the changes in protein abundance and represent an additional level of regulation. The overall independence of the structural changes to the abundance highlights the need of the cell for a fast and effective reaction to adapt to stress conditions. This can be demonstrated with the mitochondrial protein Mge1. The clogger expression renders mature Mge1 unstable. Mge1 is the mitochondrial nucleotide exchange factor of the matrix Hsp70 protein and is essential to convert Hsp70 into its ATP-bound form (Laloraya et al, 1994; Westermann et al, 1995; Slutsky-Leiderman et al, 2007). Both for the yeast Mge1 and for its human homologs GrpEL1/2, it was shown that oxidative stress conditions switch these proteins from an active soluble state into amyloid-type aggregates (Srivastava et al, 2017; Karri et al, 2019). It was proposed that this conversion serves as a safety mechanism to switch off the mitochondrial import motor upon stress conditions (Craig, 2018; Mokranjac, 2020; Michaelis et al, 2022). Our proteome-wide analysis supports this structural and thereby functional change of Mge1, suggesting that clogger induction impairs ATP consumption by the import motor. In addition, recent work demonstrated that the redistribution of mature mitochondrial Hsp70, which is involved in protein folding and import, is able to regulate the efficiency of protein import (Banerjee et al, 2022 Preprint; Michaelis et al, 2022). With our ppTPP approach, we were able to detect such maturation-specific structural changes on a proteome-wide scale, which reveals new insight about protein functions and associations. Apparently, many different cellular complexes changed predominantly on the stability rather than on the abundance level when mitochondrial protein import was inhibited. Previous studies have shown that protein thermal stability often arises from changes in posttranslational modifications or from altered binding partners of these proteins (Becher et al, 2018). The observed stability change of histones suggests that clogger induction directly or indirectly impacts on the chromatin structure in the nucleus. This is consistent with the recent observation that clogger expression arrests the cell cycle at the entry into S phase (Krämer et al, 2023).

Interestingly, we noticed a higher stability of Hsp104 upon clogger expression, which is compatible with an increased substrate binding to this disaggregase. Hsp104 and the small heat shock protein Hsp42 control the formation of protein granules in the yeast cytosol which transiently store misfolded proteins during stress conditions (Miller et al, 2015; Böckler et al, 2017; Grousl et al, 2018; Nowicka et al, 2021; Shakya et al, 2021; Xiao et al, 2021). Our observations suggest that cells form such Hsp104-bound granules also during mitoprotein-induced stress conditions, in consistence with recent observations about cytosolic MitoStore granules which transiently accommodate non-imported precursor proteins (Nowicka et al, 2021; Xiao et al, 2021; Krämer et al, 2023).

With the ppTPP strategy used in this study, we have performed a multidimensional, time-resolved, proteome-wide abundance and stability profiling during proteotoxic stress conditions on basis of the mitoprotein-induced stress response. Our insights illustrate the challenge of the cell to orchestrate a network of qualitative and quantitative changes. The ppTPP method described here is a very promising tool to study cellular responses to different types and amplitudes of stress conditions in the future.

## Materials and Methods

### Strains and growth conditions

All strains used in this study were derived from YPH499 (MATa *ura3 lys2 ade2 trp1 his3 leu2 arg4*) (Sikorski & Hieter, 1989) and were grown at 30°C in minimal synthetic respiratory medium (SLac) containing 0.67% (wt/vol) yeast nitrogen base and 2% lactate as carbon source. To induce the clogger from the *GAL1* promoter, cells were shifted to an SLac medium containing 0.5% galactose.

## Growth assays

Growth curves were performed in a 96-well plate using the automated ELx808 Absorbance Microplate Reader (BioTek). The growth curves started in the SLac medium without or with 0.5% galactose at $OD_{600}$ 0.1 and the $OD_{600}$ was measured every 10 min for 72 h at 30°C. The experiment was performed in triplicates and the mean was calculated and plotted in R. SD for every measurement is shown.

For drop dilution assays, an $OD_{600}$ of 1 was harvested from cultures grown under noninducing conditions during the exponential growth phase, washed with sterile water, and a serial 1:10 dilution was done. From each dilution, 3 $\mu$l were dropped on the SLac plates containing (inducing) or lacking (noninducing) 0.5% galactose, followed by incubation at 30°C. The growth was documented after different days.

## Pulsed SILAC

Yeast culture of an arginine and lysine auxotroph yeast strain YPH499 $\Delta arg4$ with clogger or cytosolic DHFR plasmid was grown in synthetic minimal medium containing 2% lactate as carbon source and "light" isotopes of arginine (Arg0, $^{12}C_6/^{14}N_4$) and lysine (Lys0, $^{12}C_6/^{14}N_2$).

In mid-logarithmic growth phase (OD 0.6–0.8), the medium was removed by centrifugation (5 min, 5,000$g$), cells were washed once in a medium not containing arginine or lysine and resuspended in a medium containing "heavy" arginine (Arg10, $^{13}C_6/^{15}N_4$) and lysine (Lys8, $^{13}C_6/^{15}N_2$) plus 0.5% Gal to induce the clogger or the cytosolic DHFR control. Samples of 50 $OD_{600}$*ml were collected by centrifugation (5 min, 5,000$g$) before induction (0 h) or after induction for 30, 90, and 270 min.

## TPP and sample preparation

For the TPP experiment, harvested cells from pulsed SILAC shift were used directly. Cells were washed with PBS, resuspended in the same buffer in volume equal to $OD_{600}$ 0.1, and 20 $\mu$l were aliquoted to 11 wells of a PCR plate. After centrifugation (5 min, 4,000$g$), the plate was subjected to a temperature gradient (37°C, 37.8°C, 40.4°C, 44°C, 46.9°C, 49.8°C, 52.9°C, 55.5°C, 58.6°C, 62°C, 66.3°C) for 3 min in a PCR machine, followed by 3 min at room temperature. Cells were lysed with 30 $\mu$l lysis buffer (50 mg/ml zymolyase, 0.8% NP-40, 1× protease inhibitor, 1× phosphatase inhibitor, 0.25 U/$\mu$l benzonase, and 1 mM $MgCl_2$ in PBS) for 30 min of shaking (200 rpm) at 30°C. Three freeze–thaw cycles (freezing in liquid nitrogen, followed by 1 min at 25°C in a PCR machine, and vortexing). The plate was then centrifuged (5 min, 2,000$g$) to remove cell debris, and the supernatant was filtered at 500$g$ for 5 min at 4°C through a 0.45-$\mu$m 96-well filter plate to remove protein aggregates. 4 $\mu$l of the flow-through were subjected to a protein determination (Pierce BCA Protein Assay #23225; Thermo Fisher Scientific). 25 $\mu$l of the remaining flow-through were mixed 1:1 with 2× sample buffer (180 mM Tris pH 6.8, 4% SDS, 20% glycerol, 0.1 g bromophenol blue). The 37°C samples were diluted with 1× sample buffer to achieve a protein concentration of 1 mg/ml. An equal volume of 1× sample buffer was added to all other samples of the same condition. Samples were stored at –80°C.

In-solution digests were performed as previously described (Hollmann et al, 2020): 10 $\mu$g of each lysate were subjected to an in-solution tryptic digest using a modified version of the Single-Pot Solid-Phase-enhanced Sample Preparation (SP3) protocol (Hughes et al, 2014; Moggridge et al, 2018). Lysates were added to Sera-Mag Beads (#4515-2105-050250, 6515-2105-050250; Thermo Fisher Scientific) in 10 $\mu$l 15% formic acid and 30 $\mu$l of ethanol. Binding of proteins was achieved by shaking for 15 min at room temperature. SDS was removed by four subsequent washes with 200 $\mu$l of 70% ethanol. Proteins were digested with 0.4 $\mu$g of sequencing grade-modified trypsin (#V5111; Promega) in 40 $\mu$l Hepes/NaOH, pH 8.4 in the presence of 1.25 mM TCEP and 5 mM chloroacetamide (#C0267; Sigma-Aldrich) overnight at room temperature. The beads were separated, washed with 10 $\mu$l of an aqueous solution of 2% DMSO, and the combined eluates were dried down.

Peptides were reconstituted in 10 $\mu$l of $H_2O$ and reacted with 80 $\mu$g of TMT10plex (#90111; Thermo Fisher Scientific) (Werner et al, 2014) label reagent dissolved in 4 $\mu$l of acetonitrile for 1 h at room temperature. Here, different conditions of one temperature point were combined in a single TMT experiment (see Fig 1A). Excess TMT reagent was quenched by the addition of 4 $\mu$l of an aqueous solution of 5% hydroxylamine (438227; Sigma-Aldrich). Peptides were mixed, subjected to a reverse phase clean-up step (OASIS HLB 96-well $\mu$Elution Plate, #186001828BA; Waters), and subjected to an off-line fractionation under high pH condition (Hughes et al, 2014).

The resulting 12 fractions were then analyzed by LC–MS/MS on an Orbitrap Fusion Lumos mass spectrometer (Thermo Fisher Scientific) as previously described (Sridharan et al, 2019). To this end, peptides were separated using an UltiMate 3000 RSLCnano system (Dionex) equipped with a trapping cartridge (Precolumn C18 Pep-Map100, 5 mm, 300 $\mu$m i.d., 5 $\mu$m, 100 Å) and an analytical column (Acclaim PepMap 100. 75 × 50 cm C18, 3 mm, 100 Å) connected to a nanospray-Flex ion source. The peptides were loaded onto the trap column at 30 $\mu$l per min using solvent A (0.1% formic acid) and eluted using a gradient from 2–40% Solvent B (0.1% formic acid in acetonitrile) over 2 h at 0.3 $\mu$l per min (all solvents were of LC–MS grade). The Orbitrap Fusion Lumos was operated in positive ion mode with a spray voltage of 2.4 kV and capillary temperature of 275°C. Full scan MS spectra with a mass range of 375–1,500 m/z were acquired in profile mode using a resolution of 120,000 (maximum fill time of 50 ms or a maximum of $4 \times 10^5$ ions [AGC] and a RF lens setting of 30%. Fragmentation was triggered for a 3-s cycle time for peptide-like features with charge states of 2–7 on the MS scan (data-dependent acquisition). Precursors were isolated using the quadrupole with a window of 0.7 m/z and fragmented with a normalized collision energy of 38. Fragment mass spectra were acquired in profile mode and a resolution of 30,000 in the profile mode. Maximum fill time was set to 64 ms or an AGC target of $1 \times 10^5$ ions). The dynamic exclusion was set to 45 s.

Acquired data were analyzed using IsobarQuant (Franken et al, 2015) and Mascot V2.4 (Matrix Science) using a reverse UniProt FASTA Saccharomyces_cerrevesiae_database (UP000002311) including common contaminants. To distinguish between newly synthesized SILAC-labelled (heavy) and nonlabelled (light) proteins, two separate Mascot searches were conducted. (1) The following modifications were taken into account for the identification of mature (light, pre induction) proteins: Carbamidomethyl

(C, fixed), TMT10plex (K, fixed), acetyl (N-term, variable), oxidation (M, variable), and TMT10plex (N-term, variable). (2) For the analysis of peptides derived from newly synthesized isotope-labelled proteins (heavy), the following modifications were considered as previously described (Määttä et al, 2020): Carbamidomethyl (C, fixed), label: 13C(6)15N(4) (R, fixed), TMT10plexSILAC (K, fixed; composition: 13C(10) 15N(3)C(2)H(20)N(-1)O(2)), acetyl (protein N-term, variable), oxidation (M, variable), TMT10plex (N-term, variable).

The mass error tolerance for full scan MS spectra was set to 10 ppm, and for MS/MS spectra, to 0.02 D. A maximum of two missed cleavages were allowed. A minimum of two unique peptides with a peptide length of at least seven amino acids and a false discovery rate below 0.01 were required on the peptide and protein levels (Savitski et al, 2015).

The raw output files of IsobarQuant (protein.txt – files) were processed using the R programming language (ISBN 3-900051-07-0). Only proteins that were quantified with at least two unique peptides were considered for the analysis. Moreover, only proteins which were identified in two out of three mass spec runs in one of the lowest temperatures (37°C or 37.8°C), minimum 5 out of 11 temperatures, and identified in at least eight mass spec runs over all temperatures were kept for the analysis. Raw reporter ion signals ("signal_sum" columns) were first cleaned for batch effects using limma (Ritchie et al, 2015) and further normalized using vsn, variance stabilization normalization (Huber et al, 2002). Different normalization coefficients were estimated for each temperature and heavy or light experiments. Abundance and stability scores were calculated as indicated in Mateus et al (2020). A clogger/control ratio was calculated for each temperature and SILAC label separately. The abundance score was estimated by calculating an average ratio of the first two temperatures (37°C and 37.8°C) for each replicate and SILAC label. The remaining ratios were then divided by the respective abundance average and summed up to calculate the stability score. Abundance and stability score were transformed into z-distribution using the "scale" function of R. Both scores were scaled separately for each SILAC label. To estimate the significance, abundance and stability scores for each replicate have been tested for difference using limma (Ritchie et al, 2015). The number of identifications in the different temperatures for each replicate has been used as a weight. The t-values (output of limma) were analyzed with the "fdrtool" function of the fdrtool package (Strimmer, 2008) to extract P-values and false discovery rates (fdr - q-values).

Experiments were performed in n = 3 independent biological replicates. For correlation, the Pearson method was used. R = Pearson coefficient; P-value <0.05 was considered as significant. For distributional testing two-sided Kolmogorov–Smirnoff test was used. P-values <0.05 were considered as significant. The used tests are indicated in the figure legends. For reasons of clarity, the density plots showing the distribution of the stability score were cut to concentrate on the main distribution rather than on the outlier proteins. The programming language R (https://www.r-project.org) was used to analyze the data. All figures generated were assembled in CorelDraw X7.

### Gene ontology enrichment analysis

Gene enrichment analysis was performed using GOrilla (Eden et al, 2009) based on GO terms "Biological Process" and "Cellular compartment." All quantified proteins found in the respective SILAC channel were used as background. A custom list was used for target proteins. Analysis was performed using two unranked lists of genes (target and background lists). Benjamini–Hochberg-corrected p-values were used. Only the top results with a false discovery rate <5% were considered and shown.

### Localization analysis

Localization of a protein was defined by GO-term or custom lists. The list of proteins linked to "mitochondria" was acquired using data from a previous study (Morgenstern et al, 2017). The list of proteins assigned to protein-folding proteins (GO:0006457), ubiquitin–proteasome system (GO:0000502) and 80S ribosome (GO: 0022626, or custom list filtered for "Rpl" or "Rps" proteins) were collected using genome-wide annotation for yeast, primarily based on mapping using ORF identifiers from SGD. Bioconductor package "org.Sc.sgdSGD" (Bioconductor). List of proteins assigned to the OXPHOS system was gathered using the data from Boos et al (2019). List of mitoribosomal proteins was generated using data from Desai et al (2017).

### Calculation of synthesis and degradation rates

To calculate protein synthesis rates, the increase in heavy peptide intensity over time (30′, 90′, and 270′) at 37°C corrected for batch-effects was used for analysis. For protein degradation rates, the decrease in light peptide intensity over time (0′, 30′, 90′, and 270′) at 37°C corrected for batch-effects was used for analysis. A statistical approach was used to study the protein synthesis rate (KS)/ degradation rate (KD). In this mathematical model, it is assumed that proteins are synthesized/degraded exponentially along the time course. The ΔKS/ΔKD value was calculated by determining the ratio between KS or KD of the clogger and the control strain.

### Transcriptional data

For the correlation analysis between transcription and translation, transcriptional data from a previous study were used (Boos et al, 2019). For the analysis, the data generated 270′ after induction was used.

### Presentation of ribosomal 3D models

Densities were generated based on the atomic coordinates of a S. cerevisiae 80S Ribosome (PDB-6Q8Y) (Tesina et al, 2019) by simulating densities at 7 Å resolution using the "molmap" functionality in UCSF Chimera (Pettersen et al, 2004). Simulated densities were visualized using UCSF ChimeraX (Goddard et al, 2018). Figures were assembled using Inkscape.

## Data Availability

The protein MS datasets produced in this study are available in the PRIDE database with the ProteomeXchange identifier PXD037741.

Codes used for the analyses are deposited in GitHub (https://github.com/fstein/Mitochondrial_SILAC_2DTPP).

# Supplementary Information

# Acknowledgements

We thank Andrea Trinkaus, Vera Nehr, and Sabine Knaus for technical assistance. We thank André Mateus for help with establishing the TPP protocol in yeast. We thank Timo Mühlhaus for discussions. This study was financially supported by the European Research Council (ERC 101052639 MitoCyto to JM Herrmann), the Deutsche Forschungsgemeinschaft (HE2803/10-1 and GRK2737-STRESSistance to JM Herrmann), the Landesforschungsinitiative Rheinland-Pfalz BioComp (to F Boos and JM Herrmann), the Damon Runyon Cancer Research Foundation (DRG-2461-22 to F Boos), and the Joachim Herz Stiftung (to F Boos and C Groh).

## Author Contributions

C Groh: conceptualization, data curation, formal analysis, validation, investigation, visualization, methodology, and writing—review and editing.
P Haberkant: data curation, formal analysis, visualization, methodology, and writing—review and editing.
F Stein: data curation, software, formal analysis, investigation, visualization, methodology, and writing—review and editing.
S Filbeck: formal analysis, investigation, visualization, and writing—review and editing.
S Pfeffer: validation, investigation, visualization, and writing—review and editing.
MM Savitski: conceptualization, investigation, methodology, and writing—review and editing.
F Boos: conceptualization, data curation, formal analysis, supervision, investigation, visualization, methodology, project administration, and writing—review and editing.
JM Herrmann: conceptualization, data curation, supervision, investigation, visualization, project administration, and writing—original draft, review, and editing.

## Conflict of Interest Statement

The authors declare that they have no conflict of interest.

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
