## [Reviewer comments · Life Science Alliance]

Mitochondrial dysfunction rapidly modulates the abundance and thermal stability of cellular proteins

Carina Groh, Per Haberkant, Frank Stein, Sebastian Filbeck, Stefan Pfeffer, Mikhail Savitski, Felix Boos, and Johannes Herrmann

DOI: <https://doi.org/10.26508/lsa.202201805>

Corresponding author(s): Johannes Herrmann, University of Kaiserslautern and Felix Boos, University of Kaiserslautern

Review Timeline:

Submission Date:	2022-11-04
Editorial Decision:	2022-12-20
Revision Received:	2023-01-23
Editorial Decision:	2023-02-10
Revision Received:	2023-03-03
Accepted:	2023-03-03

Scientific Editor: Novella Guidi

Transaction Report:

December 20, 2022

Re: Life Science Alliance manuscript #LSA-2022-01805-T

Johannes M Herrmann
University of Kaiserslautern
Cell Biology
Erwin-Schroedinger-Strasse 13
Kaiserslautern D-67663
Germany

Dear Dr. Herrmann,

Thank you for submitting your manuscript entitled "Mitochondrial dysfunction rapidly modulates the abundance and thermal stability of cellular proteins" to Life Science Alliance. The manuscript was assessed by expert reviewers, whose comments are appended to this letter. We invite you to submit a revised manuscript addressing the Reviewer comments.

Thank you for this interesting contribution to Life Science Alliance. We are looking forward to receiving your revised manuscript.

Sincerely,

B. MANUSCRIPT ORGANIZATION AND FORMATTING:

Reviewer #1 (Comments to the Authors (Required)):

In their manuscript entitled "Mitochondrial dysfunction rapidly modulates the abundance and thermal stability of cellular proteins", Groh et al. developed a new approach, ppTPP, to monitor changes in stability of proteins in response to induced stress. This was done by combining SILAC and two-dimensional TPP proteomics. The ppTPP assay was used here to investigate how inhibition of mitochondrial protein import, induced by expression of a clogger, impacts the cellular proteome and proteins' thermal stability. SILAC provided an additional level of analysis and enabled dissection of the effects of mitochondrial clogging on pre-existing mature proteins versus proteins synthesized during stress.

The first part of the manuscript is focused on the impact of mitoprotein-induced stress on protein abundance: 1. Analysis of ppTPP clogger data suggested a general reduction in protein synthesis with a more profound impact on mitochondrial proteins. 2. A global increase in degradation of mature proteins (synthesized prior to stress induction) was detected in clogger-expressing cells, including various mitochondrial and peroxisomal proteins. In the second part, the authors examined alterations in protein stability by analyzing the thermal stability data. A relatively high decrease in the stability of newly synthesized mitochondrial proteins demonstrated their susceptibility to inhibited import. Changes were also detected in the stability of certain chaperones and various subunits of protein complexes such as histones. Finally, stabilization of specific ribosomal subunits was shown to occur in cells expressing a clogger. This was demonstrated for both newly synthesized and already existing subunits and was suggested by the authors as a cause or consequence of translation attenuation.

A growing body of evidence suggests that the impact of mitochondrial protein import defects can be detrimental to cells. Consequences of such stress can damage mitochondria but also cause cytosolic proteotoxic stress, which burdens the cytosolic protein quality control pathways. It is therefore timely and important to reveal the toxicity of un-imported mitochondrial precursors as a whole, and the contribution of each mitochondrial protein to proteotoxicity. Therefore, this manuscript's focus on developing and employing ppTPP to reveal changes in cellular proteins upon clogging of mitochondria is very interesting and can contribute greatly to the field. In addition, the establishment of ppTPP could benefit research of other cellular stress pathway. The manuscript includes extensive high-quality data and validates previous discoveries, as well as presents new findings. In particular, the alteration in the stability of ribosomal subunits during mitoprotein-induced stress is exciting, and can suggest a potentially novel mechanism to regulate translation.

Major comments:

- My main concerns revolve around the conclusions made by the authors regarding proteins' abundance and degradation. It is unclear if the data presented support the conclusions. In particular:
 1. In page 9- "Compared to control cells, clogger expression globally reduced protein synthesis, in particular upon prolonged clogger induction (Fig. 3B, EV3A). Thus, the rates of protein synthesis (Fig. EV3B) were significantly reduced upon mitoprotein-induced stress (Fig. 3C, D)." Can this assay distinguish protein synthesis from protein degradation and aggregation? The authors should either add controls or provide an explanation to exclude the possibility that:
 - A. The newly synthesized proteins were sequestered in aggregates (and were therefore excluded from the analysis).
 - B. The newly synthesized proteins were degraded faster, due to their inability to fold properly. Such conditions could potentially be exacerbated in the clogger expressing cells that experience overall overwhelming of cellular QC mechanisms including chaperons.
 2. Same as A for data presented in page 10- "In summary, cells complement the stress-induced changes in gene expression by consistent and protein-specific regulation patterns that operate on a post-transcriptional level (Fig. 3H)."
 3. Figure 4- Is it possible that the light labeled proteins were aggregated and not degraded?

For points 1-3 above: If the authors believe that the above interpretations could be excluded by the way the assay was conducted, the text should cover how.

- The authors should comment on the fact that the abundance data was collected at ~37 degrees (or add a non-heat shock control at 30 degrees). Is it possible that under these heat shock conditions the cytosolic DHFR will impact cells differently from the mitochondrial DHFR clogger? For example, the combination of heat shock and clogger can exacerbate aggregation.

Minor comments:

1. Page 11- "Stability changes reveal clogger-induced repression in protein synthesis"- the data revealed stabilization of ribosomal subunits in response to mitoprotein-induced stress, which is interesting and novel. However, the authors did not provide evidence that link the stability changes to repression of protein synthesis. At this time, this is only a correlation and thus the conclusion should be toned down.
2. Page 12- The following sentence is unclear- please revise: "Interestingly, here even those proteins were destabilized that had been already present before clogger induction, suggesting that the stability of these proteins is directly or indirectly influenced by the import rate of proteins into mitochondria."

Reviewer #2 (Comments to the Authors (Required)):

This is a very important study that establishes the use of mass spectrometric thermal stability measurements on the scale of the entire proteome in combination with the measurements of protein abundance. The authors provide the overview of the abundance and thermal stability changes of the proteome in response to clogging the mitochondrial protein import channels. This is a nice study that constitutes the proof of concept for the use of such technology to identify potentially interested changes. Many observations concerning the changes in the thermal stability align nicely with the published data. They are also some other interesting changes that currently are not possible to explain and surely there will be a source of further research by the authors or other interested groups.

Some comments below may help to improve the overall message of this paper.

Major comments:

- Could authors discuss the behavior of the membrane proteins in this assay, and specifically the protein translocases
- Explanation of the approach is quite complicated. The terminology "pre" vs. "post" is unlucky, as it suggests that the heavy label is indicating a time course post clogging event, however, it could also be post DHFR control induction, thus without clogging. Further, one may initially assume "pre" to describe only the time point 0. Why not use 'mature' and 'nascent/de novo' or similar? Usage of the term "stable" as on p. 7 (3rd paragraph "generally stable levels") should be avoided in the context of protein abundance so not to confuse with thermal stability.
- Figure 4G shows the most promising look into temporal aspects (kinetics), however, final analysis is digital ("low/high/none") - seems underexplored
- Why did authors not calculate and plot a fold change post/pre and then show the time course, because this is what is being analyzed?
- The changes in the stability of the cytosolic ribosome proteins, especially those in the tunnel and in the close proximity of the exit suggest that the changes in translation happen in the elongation stage (Topf et al.) and not through decreasing the levels of ribosomes or translation initiation. This interpretation is worth of deeper discussion as the elongation stage is not typically considered as a stress response target.
- The authors should ensure that the original data are deposited properly for the other researchers to use.

Minor comments

- Labeling order of Pre/Post is inconsistent making it harder to follow (Figure 1, 3 anfollowing: Pre/Post, Figure 2, EV3B, EV6B: Post/Pre).
- Figure 1B and part of 2A are identical/redundant?
- Avoid overstating, e. g. in introduction (p. 5) "Strikingly", in conclusion "Differentiating from previous methods (p. 14 > it is rather 'combining previous methods')
- Avoid predictions as in "We are confident that the ppTPP method described here will proof to be [...] widely used" (p. 16, last sentence)
- Nowicka et al - this citation is consequently misspelled

Point-by-point response

Referee #1

A growing body of evidence suggests that the impact of mitochondrial protein import defects can be detrimental to cells. Consequences of such stress can damage mitochondria but also cause cytosolic proteotoxic stress, which burdens the cytosolic protein quality control pathways. It is therefore timely and important to reveal the toxicity of un-imported mitochondrial precursors as a whole, and the contribution of each mitochondrial protein to proteotoxicity. Therefore, this manuscript's focus on developing and employing ppTPP to reveal changes in cellular proteins upon clogging of mitochondria is very interesting and can contribute greatly to the field. In addition, the establishment of ppTPP could benefit research of other cellular stress pathway. **The manuscript includes extensive high-quality data and validates previous discoveries, as well as presents new findings. In particular, the alteration in the stability of ribosomal subunits during mitoprotein-induced stress is exciting, and can suggest a potentially novel mechanism to regulate translation.**

We thank the referee for her/his very positive evaluation!

Major comments:

- My main concerns revolve around the conclusions made by the authors regarding proteins' abundance and degradation. It is unclear if the data presented support the conclusions. In particular:

1. In page 9- "Compared to control cells, clogger expression globally reduced protein synthesis, in particular upon prolonged clogger induction (Fig. 3B, EV3A). Thus, the rates of protein synthesis (Fig. EV3B) were significantly reduced upon mitoprotein-induced stress (Fig. 3C, D)." Can this assay distinguish protein synthesis from protein degradation and aggregation? The authors should either add controls or provide an explanation to exclude the possibility that:

A. The newly synthesized proteins were sequestered in aggregates (and were therefore excluded from the analysis).

We agree with the referee. We precisely measure the generation of newly synthesized proteins (which are characterized by their incorporation of heavy isotopes) in the NP40-soluble fraction. Thus, NP40-insoluble proteins are not measured. We now changed the text and describe this more explicitly. However, since the accumulation of newly synthesized peptides is globally reduced in clogger-expressing cells and not restricted to specific proteins, it seems unlikely that this effect is explained by the incorporation into NP40-insoluble aggregates or the pre-mature aggregation of proteins. However, for individual proteins these processes might certainly be of relevance.

Furthermore, it should be noted that the conclusions we made on the effect of clogger expression on the generation of NP40-soluble proteins are consistent with what we observed in a previous study for the generation of urea/SDS-soluble proteins (Boos et al. 2019 Nature Cell Biology 21, 442-451).

Following the recommendation of the referee, we rewrote several parts of the study and discuss the potential effect of NP40-insoluble aggregates. We appreciate this suggestion of the referee to be more cautious in our interpretation.

B. The newly synthesized proteins were degraded faster, due to their inability to fold properly. Such conditions could potentially be exacerbated in the clogger expressing cells that experience overall overwhelming of cellular QC mechanisms including chaperons.

We also agree in this point. If proteins are degraded during or directly after translation we would interpret this as being not synthesized. We also mention this explicitly in the text. This aspect was already discussed in the context of mitochondrial proteins for which we observed a strong reduction in the abundance of heavy peptides that were detected in the presence of the clogger. But we extended now our discussion as this clogger-induced effect might arise from reduced transcription/translation or increased degradation of nascent polypeptides.

2. Same as A for data presented in page 10- "In summary, cells complement the stress-induced changes in gene expression by consistent and protein-specific regulation patterns that operate on a post-transcriptional level (Fig. 3H)."

Here we compare the changes on the RNA level as detected by RNA sequencing to the levels in protein. Therefore, we feel that the sentence is correct as it stands because we compare the changes on a transcriptional level (leading to changes in the amount of RNA) to changes on a post-transcriptional level (leading to changes in the amount of protein). We did not differentiate between effects on translation, aggregation or degradation of proteins here, thereby inherently accounting for the caveat raised by the reviewer.

3. Figure 4- Is it possible that the light labeled proteins were aggregated and not degraded?

As described above, it is possible because we do not measure NP40-insoluble peptides. We describe this now in more detail. We are grateful for this comment as we indeed recently observed that non-imported mitochondrial proteins can accumulate in cytosolic aggregates. The formation of these aggregates depends on specific factors such as the small heat shock protein Hsp42. It will be interesting to compare the overlap of proteins that are reduced in the NP40 fractions upon clogger expression with those that are included into cytosolic aggregates. However, we feel that such additional datasets would be beyond this study.

We are now discussing this aspect specifically.

For points 1-3 above: If the authors believe that the above interpretations could be excluded by the way the assay was conducted, the text should cover how.

We followed the recommendation of the referee to add a more comprehensive interpretation of our results.

- The authors should comment on the fact that the abundance data was collected at ~37 degrees (or add a non-heat shock control at 30 degrees). Is it possible that under these heat shock conditions the cytosolic DHFR will impact cells differently from the mitochondrial DHFR clogger? For example, the combination of heat shock and clogger can exacerbate aggregation.

For all samples shown in the study, the cells were grown at 30°C to avoid the influence of a heat shock response. Indeed, high temperature would exacerbate aggregation. The temperature ramp before NP40 lysis was only for a very short time (3 min). Therefore, we do not expect that a heat shock response would alter the levels of proteins. The temperature ramp used had been optimized previously to cover the individual 'melting curve' of proteins which for most proteins is in the range between 48 and 60°C. Basically all cellular proteins maintain their NP40 solubility upon incubation for this short time at

temperatures below 40°C and thus, data points below that temperature do not add more information about the thermal stability of proteins, which is why these were omitted in our experiment.

Minor comments:

1. Page 11- "Stability changes reveal clogger-induced repression in protein synthesis"- the data revealed stabilization of ribosomal subunits in response to mitoprotein-induced stress, which is interesting and novel. However, the authors did not provide evidence that link the stability changes to repression of protein synthesis. At this time, this is only a correlation and thus the conclusion should be toned down.

The sentence was indeed misleading and we changed it now. We found the stability of ribosomal proteins to be increased upon clogger induction. This is interesting as clogger induction reduces cytosolic protein synthesis. Thus, reduced translation and structural changes of the 80S ribosome 'correlate', as pointed out by the reviewer. Of course, our data do not establish a causality, even though it is likely that structural changes of the ribosome are directly linked to their function and, hence, to translation rates. To avoid overinterpretation, we rephrased the title of the section to: 'Clogger expression changes the thermal stability of proteins of the ribosomal tunnel exit'.

The clogger-induced reduction of protein synthesis was experimentally shown before (Boos et al. 2019 Nature Cell Biology 21, 442-451). We show the respective figure panel of that previous study here again for inspection by the referee.

Figure 1. Mitoprotein-induced stress inhibits cytosolic protein synthesis. Radiolabelling of newly synthesized proteins showed a strong reduction of translation rates 3 h after clogger induction. This data item is shown as Supplementary Figure 4a in Boos et al., 2019.

2. Page 12- The following sentence is unclear- please revise: "Interestingly, here even those proteins were destabilized that had been already present before clogger induction, suggesting that the stability of these proteins is directly or indirectly influenced by the import rate of proteins into mitochondria."

We rephrased the sentence.

Reviewer #2

This is a very important study that establishes the use of mass spectrometric thermal stability measurements on the scale of the entire proteome in combination with the measurements of protein abundance. The authors provide the overview of the abundance and thermal stability changes of the proteome in response to clogging the mitochondrial protein import channels.

This is a nice study that constitutes the proof of concept for the use of such technology to identify potentially interested changes. Many observations concerning the changes in the thermal stability align nicely with the published data. They are also some other interesting changes that currently are not possible to explain and surely there will be a source of further research by the authors or other interested groups.

Some comments below may help to improve the overall message of this paper.

We thank the referee for her/his very positive evaluation!

Major comments:

- Could authors discuss the behavior of the membrane proteins in this assay, and specifically the protein translocases

This is an interesting aspect. In terms of abundance, we actually found many membrane proteins to be changed considerably. For example, we found Atm1 or Cox15 among the mitochondrial proteins that were most severely depleted upon clogger expression. Atm1 and Cox15 are among the most hydrophobic mitochondrial proteins. However, when we analyzed thermal stability, we found hardly any really hydrophobic mitochondrial proteins to be considerably changed upon clogger expression (compare Fig. 6F, G). Most of these membrane proteins had just single transmembrane domains or were beta-barrel proteins such as Tom40 or Por1. It is well possible that deeply membrane-embedded proteins are less prone to clogger-induced structural changes than soluble proteins. We mention this now in the text.

- Explanation of the approach is quite complicated. The terminology "pre" vs. "post" is unlucky, as it suggests that the heavy label is indicating a time course post clogging event, however, it could also be post DHFR control induction, thus without clogging. Further, one may initially assume "pre" to describe only the time point 0. Why not use 'mature' and 'nascent/de novo' or similar? Usage of the term "stable" as on p. 7 (3rd paragraph "generally stable levels") should be avoided in the context of protein abundance so not to confuse with thermal stability.

The terms mature and nascent have been indeed used in the past (e.g. Savitski *et al.* 2018. Cell 173, 260-274). However, we felt that these terms are even more misleading as nascent typically refers to proteins during the process of their synthesis, thus to growing polypeptide chains. Following also the line of referee #1, we felt that such a nomenclature would be dangerous as we are measuring the accumulation of proteins that are synthesized before or after clogger induction; however, their levels are not solely influenced by synthesis rates but also by degradation and aggregation. We improved our discussion to explain this in more detail.

For example, we added a sentence into the introduction stating: 'We refer here to these mature and newly synthesized proteins as 'pre' and 'post', respectively, owing to their synthesis before and after clogger induction and label shift.'

- Figure 4G shows the most promising look into temporal aspects (kinetics), however, final analysis is digital ("low/high/none") - seems underexplored. Why did authors not calculate and plot a fold change post/pre and then show the time course, because this is what is being analyzed?

In our previous paper (Boos et al. 2019 Nature Cell Biology 21, 442-451) we analyzed the temporal aspects of the transcriptomics and proteomics data in depth. Dynamic changes are much more pronounced on the transcriptome level and translate only slowly into proteome changes. Nevertheless, we see considerable time-dependent reactions also for the changes in thermal stability. In order to visualize these changes more clearly we now provide an additional supplemental dataset with pdfs of all proteins identified in this study. So, it will be easy for readers to scroll through them and look for their specific proteins of interest and their individual time-dependent changes upon clogger expression.

- The changes in the stability of the cytosolic ribosome proteins, especially those in the tunnel and in the close proximity of the exit suggest that the changes in translation happen in the elongation stage (Topf et al.) and not through decreasing the levels of ribosomes or translation initiation. This interpretation is worth of deeper discussion as the elongation stage is not typically considered as a stress response target.

We discuss the effects on the translation machinery now in more depth also mentioning the excellent previous studies by the Chacinska lab. Their observations are in full agreement with the data shown in this study. However, after careful consideration, we decided not to include a discussion at which stage (initiation/elongation) translation is affected. We agree with the reviewer that our data suggest that elongation could be altered. Given that this is a paradigm shift in the field of translation regulation upon stress, we feel that it would be preliminary to make any statement on this topic based on our dataset. It is certainly an exciting possibility that warrants further in-depth investigation.

- The authors should ensure that the original data are deposited properly for the other researchers to use.

We provide the dataset in an easily accessible spreadsheet format in the supplement, and uploaded the entire datasets to repositories from which they can be freely downloaded and used by others. Information on how to access the data are provided in the 'Data Availability' paragraph.

Minor comments

- Labeling order of Pre/Post is inconsistent making it harder to follow (Figure 1, 3 anfollowing: Pre/Post, Figure 2, EV3B, EV6B: Post/Pre).

We changed the respective figures and followed the Pre/Post order throughout the study.

- Figure 1B and part of 2A are identical/redundant?

We removed the graphical explanation from Fig. 2A as suggested.

- Avoid overstating, e. g. in introduction (p. 5) "Strikingly", in conclusion "Differentiating from previous methods (p. 14 > it is rather 'combining previous methods')

We changed the text as suggested. 'Strikingly' was replaced in the abstract and the introduction. The conclusion was also changed as suggested by the referee.

- Avoid predictions as in "We are confident that the ppTPP method described here will proof to be [...] widely used" (p. 16, last sentence)

We changed the text as suggested.

- Nowicka et al - this citation is consequently misspelled

We thank the reviewer for pointing this out. We corrected the reference and apologize for the misspelling.

February 10, 2023

RE: Life Science Alliance Manuscript #LSA-2022-01805-TR

Dr. Johannes M Herrmann
University of Kaiserslautern
Cell Biology
Erwin-Schroedinger-Strasse 13
Kaiserslautern D-67663
Germany

Dear Dr. Herrmann,

Thank you for submitting your revised manuscript entitled "Mitochondrial dysfunction rapidly modulates the abundance and thermal stability of cellular proteins". We would be happy to publish your paper in Life Science Alliance pending final revisions necessary to meet our formatting guidelines.

- please address Reviewer 2's remaining points
- please add ORCID ID for secondary corresponding author-they should have received instructions on how to do so
- please add a conflict of interest statement to the main manuscript text
- please rename your EV figures as supplementary figures; please adjust the figure callouts in the text accordingly
- the PRIDE dataset should be made publicly available at this point, and you can remove the Reviewer access details from the Data Availability statement
- there is a Related Manuscript File that appears to be a combination of other figure panels. Please either remove this file upon re-submission, or let us know what it is meant to be.

A. FINAL FILES:

B. MANUSCRIPT ORGANIZATION AND FORMATTING:

Sincerely,

Reviewer #1 (Comments to the Authors (Required)):

The authors adequately addressed my comments and provided additional clarifications. The method established in this study as well as the data obtained using it will greatly contribute to the broad scientific community. The manuscript will be of great interest to the Life Science Alliance journal readers.

Reviewer #2 (Comments to the Authors (Required)):

The authors addressed the comments in a sufficient way.

A few comments below are for the authors consideration to present their findings and promote the usage of the original datasets by the others.

It is laudable the authors now include quantitative data for all 2804 proteins in a in supplement dataset S1. Upon inspection of highlighted examples, a partially incomplete data (e. g. only 3/33 datapoints for Atm1, 7/33 datapoints for Cox15 measured) was noticed.

In addition, we want to point out that for example for Tom40 (that is listed as a protein that in the light/pre condition "showed most severe reduction in stability scores upon clogger expression" (Fig. 6F,G)), we could not visually observe any differences. In the absence of quantitative values other than the figures in dataset S1, we were not able to calculate stability scores ourselves. The paper and the dataset is very valuable, even without full set of replicates for some proteins, but the authors should be sensitized to base their examples and conclusions on high-quality reproducible data.

Minor points:

- Supplementary Dataset S1 would greatly benefit from a separate caption explaining each panel.
- Illegible glyphs in the PRIDE submission text.

All other comments are sufficiently addressed.

March 3, 2023

RE: Life Science Alliance Manuscript #LSA-2022-01805-TRR

Dr. Johannes M Herrmann
University of Kaiserslautern
Cell Biology
Erwin-Schroedinger-Strasse 13
Kaiserslautern D-67663
Germany

Dear Dr. Herrmann,

Thank you for submitting your Research Article entitled "Mitochondrial dysfunction rapidly modulates the abundance and thermal stability of cellular proteins". It is a pleasure to let you know that your manuscript is now accepted for publication in Life Science Alliance. Congratulations on this interesting work.

DISTRIBUTION OF MATERIALS:

Again, congratulations on a very nice paper. I hope you found the review process to be constructive and are pleased with how the manuscript was handled editorially. We look forward to future exciting submissions from your lab.

Sincerely,
